# Unveiling Misconceptions among Small-Scale Farmers Regarding Ticks and Tick-Borne Diseases in Balochistan, Pakistan

**DOI:** 10.3390/vetsci11100497

**Published:** 2024-10-12

**Authors:** Zafar Ullah, Mehran Khan, Iram Liaqat, Kashif Kamran, Abdulaziz Alouffi, Mashal M. Almutairi, Tetsuya Tanaka, Abid Ali

**Affiliations:** 1Microbiology Lab, Department of Zoology, Government College University, Lahore 54000, Punjab, Pakistan; 2Department of Zoology, University of Balochistan Quetta, Quetta 87300, Balochistan, Pakistan; 3Department of Zoology, Abdul Wali Khan University Mardan, Mardan 23200, Khyber Pakhtunkhwa, Pakistan; 4King Abdulaziz City for Science and Technology, Riyadh 12354, Saudi Arabia; 5Department of Pharmacology and Toxicology, College of Pharmacy, King Saud University, Riyadh 11451, Saudi Arabia; 6Laboratory of Animal Microbiology, Graduate School of Agricultural Science/Faculty of Agriculture, Tohoku University, 468-1 Aramaki Aza Aoba, Aoba-ku, Sendai 980-8572, Japan

**Keywords:** ticks, tick-borne diseases, farmers, grazing animals, knowledge, attitude, practices

## Abstract

**Simple Summary:**

Ticks and tick-borne diseases (TBDs) significantly affect the health and production of small ruminants, particularly among resource-poor and small-scale farmers in the Balochistan province of Pakistan. This study surveyed 153 farmers across seven districts to assess their knowledge, attitudes, and practices using a KAP survey. The results revealed a significantly low level of awareness among farmers about the impact of climate change and the economic effects of ticks on animal health. The key preventive measures, such as proper acaricide use and wearing protective clothing, were often neglected, increasing the risk of TBDs. The roles of the government, non-government organizations, veterinary doctors, and local communities are essential to implement effective awareness and education programs to address these gaps.

**Abstract:**

Ticks and tick-borne diseases (TBDs) pose potential health threats to small-scale farmers of grazing animals in the upper highlands of Balochistan, Pakistan. This study was conducted based on a questionnaire survey involving 153 farmers of grazing animals in seven districts to access their knowledge, attitudes, and practices regarding ticks and TBDs. Odds ratios and 95% confidence intervals, based on Fisher’s test, were used to assess risk factors for determining preventive measures. The findings revealed a low level of knowledge among the participants. For instance, there was a lack of awareness of the effects of climate change and the economic impact of ticks on animal health. The essential precautions, such as the non-indiscriminate use of acaricides, wearing dark-colored clothing, and limiting children’s interaction with grazing animals, were often overlooked. However, the farmers had a positive attitude towards tick control, but they mostly relied on the knowledge of local communities. The neglect of such measures places these farmers and their children at risk of contracting TBDs. This study also indicates minimal involvement from the government in educating farmers and controlling ticks. The role of stakeholders, including the government, non-governmental organizations, veterinary doctors, and local farmer communities, is crucial to address these issues and to implement effective training programs that address misconceptions about ticks and TBDs. Overall, this study highlights the importance of implementing awareness and education programs to address the misconceptions about ticks and TBDs among farmers.

## 1. Introduction

Located in South Asia, Pakistan ranks as the 33rd largest country by area in the world. It has a population of 241.5 million, which makes it the world’s sixth-most-populous country [1]. The country has been recognized as an agricultural nation, where the livestock sector contributes 60.84% to the value of agriculture and 14% to the total Gross Domestic Product (GDP) [2]. Due to the increase in population, urbanization, and deforestation, a significant increase in food demand has been recorded over recent decades. The livestock sector has become the fastest growing part of the agricultural industry, and this sector fulfills the food demand of the country [3]. Balochistan is the largest province of Pakistan by area, covering 44% of the land. The economy of this province primarily depends on livestock farming, which accounts for over 50% of the GDP [4].

The ‘One Health’ concept has not been widely practiced in developing countries, including Pakistan. Compared to other countries, Pakistan has had a high prevalence of infectious diseases and hazardous biological materials in the last decade, which have significantly affected the environment and human and animal welfare [5]. Infectious diseases, including chickenpox, scabies, measles, tuberculosis, and leishmaniasis, pose significant challenges to the livestock farmers of this region [5,6]. Moreover, tick-borne infections, including theileriosis, babesiosis, and Crimean–Congo hemorrhagic fever (CCHF) [7], have been reported in this region [7,8,9]. More than 40 species of tick have been identified in Pakistan [8]; however, ticks of the genus *Hyalomma* are the most prevalent in the region [9,10]. Ticks and tick-borne diseases (TBDs) are also causing parasitological issues among the rural farmers of Balochistan who are raising small ruminants, particularly sheep and goats [11,12,13]. Interactions between humans, animals, and their environment provide certain opportunities for pathogens to be transferred and spread in any direction [14,15]. For instance, CCHF led to the infection and subsequent hospitalization of 12 healthcare professionals in the Quetta district of Pakistan in 2023 [16]. In Pakistan, effective strategies to control ticks and TBDs in sheep and goats are limited [17].

Ticks play a significant role in the spread of numerous zoonotic TBDs diseases, as they carry pathogens like protozoans, bacteria, and viruses [18,19,20]. Tick-borne diseases in Pakistan have not been properly studied, and the incidence of these diseases is on the rise due to occupational health threats, a lack of vaccination, and the poor knowledge of farmers [9]. Several factors are responsible for the spread of ticks among domestic and wild animals, as well as humans. For instance, the expansion and spatial growth of urban areas promote the population expansion of ticks [21]. Another significant factor is the unregulated trade of animals, which escalates the risk of tick proliferation, and this is not necessarily under veterinary control [22]. Poverty, global environmental changes, a lack of political will, social inequality, and regional conflicts play significant roles in the emergence of zoonotic diseases [23,24].

Other studies have frequently reported that outdoor workers, particularly farmers, are at high risk of exposure to zoonotic diseases, including tick-borne zoonoses, due to their occupation. Knowledge, attitude, and practice (KAP) studies may contribute to minimizing farmers’ knowledge gaps and misconceptions regarding ticks and TBDs, which, in turn, can inform the development of targeted interventions and educational programs to overcome these risks [25,26,27]. For example, a KAP study conducted in the Netherlands revealed that citizens do not wear long-sleeved protective clothing, especially on hot days, and are also hesitant to apply insect repellent to the skin to prevent Lyme disease due to the risks of chemical exposure [28]. In Pakistan, only a few KAP studies have been conducted, which are not sufficient in determining the level of knowledge and practices among farmers [10,26,27,29,30]. Misperceptions and poor spatial and biological knowledge about ticks and TBDs among farmers are common in Balochistan [9,10]. This low level of knowledge regarding ticks and their associated risks could lead to inconsistent management practices [31]. Furthermore, inadequate farming practices addressing tick infestations can potentially lead to a disease burden and considerable economic loss [32]. This could further amplify the possibility of zoonotic TBD transmission and environmental contamination [30]. Addressing these challenges requires the coordinated efforts of veterinary doctors, farming communities, and local governments, which can be determined after the successful completion of KAP surveys.

In Balochistan, only a few articles have been published concerning ticks and TBDs [9,12,33,34]. Only a handful of them address farmers’ knowledge, while the remaining ones do not cover attitudes and practices, particularly for grazing animals. Therefore, the objectives of this study were to assess current knowledge, attitudes, and practices concerning ticks and TBDs by surveying farmers with grazing animals in seven districts in Balochistan using a socio-behavioral tool: a KAP survey.

## 2. Material and Methods

### 2.1. Study Area

Pakistan is administratively divided into four provinces. Balochistan is the largest province and the most neglected in terms of parasitological issues. The KAP survey was conducted in seven districts of Balochistan including Loralai, Musakhail, Nushki, Pishin, Quetta, Sherani, and Zhob (Figure 1). These districts were selected because grazing animals are more prevalent in these districts of Balochistan. Most of the farmers live modest lives and rely on raising sheep and goats for their livelihood. Previously, these districts had not reported issues regarding ticks and TBDs in grazing animals. Additionally, livestock extension centers and veterinary hospitals are not available in every district. The climate in these districts is arid, which supports livestock production.

### 2.2. Study Sample

All information was collected with the consent of the respondents. Respondents were selected based on their involvement with grazing animals. Initially, 185 farmers were recruited for the survey, out of which 153 provided oral and written consent to participate (Figure 2). Only 12 farmers were excluded from the survey due to the age restriction, specifically those under 18 years of age who owned farmhouses.

Verbal consent was obtained from all respondents after they were informed about the purpose of the study and their voluntary participation. They were assured that their identities would not be disclosed without their explicit consent. This study was based on random clusters of farmers recommended by the local farmer community, because their data is not registered with the Animal Husbandry Department of Balochistan, Pakistan. The use of email and internet facilities is either limited or not accessible to farmers in the study region. All questions were administered via face-to-face interviews to ensure a high response rate and accurate data collection. The purpose of this approach was to increase the reach of the survey and gather diverse perspectives on the KAP-based questionnaire. A small number of respondents (*n* = 35) also took part in the post-survey training. We actively recruited from January 2024 to June 2024, but the survey remained accessible until 30 July 2024.

### 2.3. Survey Instrument—Questionnaire

A cross-sectional questionnaire-based survey assessing KAP was conducted among farmers who grazed animals, particularly sheep and goats, from January 2024 to June 2024. Respondents were presented with the version of the survey that corresponded to the language they understood, i.e., Urdu (the national language of Pakistan), and the final survey was translated into English.

To enhance respondent understanding and ensure accurate questionnaire responses for the KAP survey, key terms were explained to each respondent prior to sharing the questionnaire. These included TBDs (diseases transmitted by ticks to animals and humans), tick-borne pathogens (protozoans, bacteria, or viruses spread by ticks), acaricides (chemicals used to eliminate ticks), antibiotics (chemical drugs used to treat bacterial infections caused by tick-borne pathogens) and vaccines (medicines that can prevent TBDs by boosting immunity). These clarifications helped to bridge knowledge gaps and improve the accuracy of the responses.

The original questionnaire (Appendix A) was developed using previously applied questionnaires on similar topics as a reference in Table 1. This table provides a structured guideline to understand respondents’ knowledge, attitudes, and practices related to ticks and TBDs as reported in previous studies, to further enhance KAP-related activities on their farms for better livestock management. In this table, two labels are provided: the first column is labeled ‘response action’, representing the expected answers in the form of ‘Yes’ or ‘I agree’ from farmers, and the second column is labeled ‘suggested actions’, offering recommendations for controlling ticks and TBDs based on insights from published articles. The initial draft of the questionnaire was shared with three small animal veterinary doctors and two university professors specializing in veterinary parasitology. This validation process ensured that the questions were relevant to the target farmers. Following validation, the questionnaire was administered as a trial to five participants. Minor modifications were made based on the feedback received from these experts to remove the ambiguities in the KAP survey. The final draft of the questionnaire was divided into four sections: (a) Socio-demographic: This section included age, marital status, ethnicity, citizenship, educational qualification, animal handling experience, monthly family income, and presence of pets along with grazing animals; (b) Knowledge base: This section covered questions related to ticks and TBDs in Balochistan; (c) Attitude: This section covered questions related to ticks and TBDs and their impact on grazing animal health farmers; and (d) Practices: This included preventive measures to reduce exposure to ticks and TBDs. Gender was not selected because tribal women are mainly confined to domestic affairs and are not allowed to herd animals. Questions on marital status, ethnicity, and urbanicity were dichotomized, while the remaining demographic questions were divided into three or four categories. The questionnaire consisted of two A4-size papers containing a total of 34 questions. Page two consisted of 28 questions related to KAP. Knowledge-based questions (*n* = 10) were assessed based on three primary categories: ‘I agree’, ‘I disagree’, and ‘I don’t know’. The response option i.e., ‘I agree’ means that the respondent confirmed the knowledge statement provided in the question, ‘I do not know’ indicates a lack of knowledge regarding the statement, and ‘I do not agree’ means that the respondent did not agree with the statement. In contrast, ‘Yes’ or ‘No’ options were provided in the attitude and practices sections to simplify the response of respondents because these sections focus on specific behaviors, which makes the binary choice more appropriate.

Questions related to attitudes (*n* = 9) and practices (*n* = 9) were recorded with ‘yes’ or ‘no’ response options. Each farmer’s survey was sealed in an envelope. It took about 20–30 min to complete the survey. We administered the questionnaire survey according to the farmers’ availability, ensuring that it was conducted on the days and times they suggested for optimal completion. A brochure and flyer about ticks and TBDs were provided as incentives in Urdu, which also supplemented their knowledge. A brochure sample designed in Urdu is attached as Appendix A.

### 2.4. Statistical Analysis

Descriptive statistics, including frequency and percentage, were used to illustrate the distribution of demographic responses using z-score in the Paleontological Statistics Software (PAST, version 4.12, https://past.en.lo4d.com/, accessed on 8 May 2024). Results for the knowledge, attitude, and practices sections were presented in graphical form using Microsoft Excel 2019^®^. To calculate percentage distribution scores for knowledge, attitude, and practices, one point was assigned if the respondent’s answer was ‘I agree’ for the knowledge section and ‘Yes’ for the attitude and practices sections. Initial risk factor assessment involved univariate analysis (unadjusted odd ratio with 95% confidence limits) applied to a logistic regression model using the Fisher exact test or *Χ*^2^ test to determine the individual impact of each selected factor for tick dispersal using Epi Info™ software (version 7.2, https://www.cdc.gov/epiinfo/index.html, accessed on 11 July 2024). Odds ratios exceeding a value of 1 were considered for multivariate analysis (adjusted odd ratios with 95% confidence limits) to assess the combined effect while controlling for potential confounders.

## 3. Results

### 3.1. Socio-Demographic Characteristics of Respondents

Most respondents in the survey were married (*n* = 137, 89.55%, *p <* 0.33) and were primarily between the ages of 25 and 45 (*n* = 65, 42.48%, *p <* 0.42) (Table 2). Most of the respondents had more than 15 years (*n* = 58, 37.91%) of grazing animal experience, and their monthly income ranged between USD 200 and 300 (*n* = 101, 68.24%, *p <* 0.18). In terms of livestock, most respondents raised goats (59.52%, *p <* 0.16), followed by sheep (31.85%) and cattle (4.12%). A few farmers also raised buffaloes (*n* = 45) and horses (*n* = 3). However, none of them had camels on their farms. Most respondents owned dogs as pets (*n* = 72, 86.75%, *p <* 0.34), while none of them reported having rabbits as pets. A small proportion of respondents (13.25%) also noted the presence of street cats, although these were not domesticated.

### 3.2. Response of Knowledge, Attitude and Practices

Most respondents (*n* = 139, 84.31%) agreed that ticks were present on their animals. However, they could not differentiate ticks from insects, and most of them (*n* = 95, 62.5%) were unaware that ticks can transmit TBDs (Figure 3). Knowledge about the life cycle of ticks was low (*n* = 131, 85.62%), and they could classify ticks based on their size. They agreed (*n* = 131, 86.18%) that ticks can jump onto the host for attachment. More than half (*n* = 82, 53.94%) agreed that climate change can impact the tick population with seasonal migration of grazing animals. Around 50% (*n* = 76) of respondents disagreed that ticks could pose any economic threat to grazing animals. Most of them (*n* = 104, 68.42%) did not know about the role of antibiotics in curing TBDs; however, they were aware of the role of effective vaccines (*n* = 83, 54.24%). A significant number of respondents (*n* = 129, 84.86%) agreed that regular tick checks can prevent ticks and TBDs.

Most of the farmers showed a positive attitude towards the control of tick infestation and consider TBDs a serious health problem (*n* = 98, 64.05%). More than half the respondents expressed concerns about tick bites and tick-borne infection (*n* = 113, 55%, Figure 4). Most of the respondents showed a positive attitude towards tick control, with spraying being perceived as the most preferred method (*n* = 134, 87.58%). The majority of respondents preferred to move their animals to the highlands during summer (*n* = 116, 75.81%). Most respondents (*n* = 121, 79.08%) were concerned about the health of grazing animals and provided medical treatment in case of illness. However, a negative attitude prevailed among respondents regarding the sale of infested animals in the market (*n* = 110, 71.89%). Farmers expressed strong beliefs in providing clean water and sufficient food (*n* = 147, 96.07%) and demonstrated a proactive attitude by separating infested animals from healthy animals during grazing (*n* = 82, 53.59%). A significant number of respondents allowed their children to interact with grazing animals.

Most respondents (*n* = 121, 79.08%) had not attended tick control training due to the limited availability of such programs provided by government or non-government organizations (NGOs) (Figure 5). Therefore, respondents mostly relied on their local farming communities (*n* = 139, 90.84%) to exchange knowledge on ticks and TBDs. Most respondents (*n* = 122, 79.73%) indiscriminately used acaricides, often disregarding the manufacturing guidelines. The government does not provide sufficient subsidies for acaricides (*n* = 148, 96.72%). Typical habits (*n* = 145, 94.77%) of ticks were not recognized for grazing animals. Limited precautionary measures, such as tucking pants into socks (*n* = 118, 77.12%) or wearing light-colored clothing (*n* = 124, 81.08%), were adopted. Most of the respondents (*n* = 131, 85.62%) neither checked their bodies for the presence of ticks after returning from fields (*n* = 101, 66.01%) nor took showers afterward.

The highest knowledge scores were recorded in the Pishin district, while the highest attitudes and practices scores were observed in the Musakhail district. Zhob district had the lowest scores in all three parameters of KAP (Figure 6).

The results of the logistic regression model using the Fisher exact test indicated that not choosing dark-colored clothing during grazing hours significantly affected tick infestation rates (Table 3). The indiscriminate use of acaricides without adhering to the manufacturer’s guidelines can promote tick resistance. Factors such as body inspection after returning from grazing [OR = 1.14 (0.4–2.75), *p <* 0.82], visits to known tick habitats along with grazing animals [OR = 1.02 (0.19–5.31), *p <* 0.65] and the impact of climate change on tick distribution [9.57 (OR = 4.54–20.19), *p <* 0.001] were found to have odds ratios above 1. Adjustments from a univariate model to a multivariate model could suggest better preventive measures. For instance, factors like clothing choice, climate change [OR = 0.75 (0.38–1.48), *p <* 0.29], and visits to known tick habits [OR = 0.20 (0.01–0.05), *p <* 0.31] were recorded as significant in the univariate model, indicating an increased risk of ticks and TBDs. Changes in these values could significantly reduce the risk factors of tick infestation and TBDs transmission.

## 4. Discussion

The current study suggests that most of the respondents had significant experience in animal husbandry, with over ten years of involvement in grazing practices. Incorrect dilution and failure to follow the manufacturer’s guidelines for acaricide use can contribute to the development of acaricide resistance [47]. Education and income levels significantly influence the knowledge and attitudes of farmers [29]. In our study, dogs were commonly kept alongside grazing animals, which is consistent with previous studies that dogs are used to guard animals during grazing [48], while cats are primarily kept to entertain children [49]. However, these pet animals were not regularly checked for the presence of ticks and thus can serve as sources to potentially transfer ticks to humans. Pet animals for the presence of ticks have been studied in an epidemiological study in Pakistan [30], highlighting the importance of these animals, as they can become a source of reverse zoonosis [50].

Most respondents recognized the presence of ticks on their grazing animals. This attitude demonstrates the importance of understanding how ticks affect these animals [30]. However, our survey respondents struggled to distinguish ticks from insects and could not differentiate between tick developmental stages [35,51]. Grazing animals were regularly checked for the presence of ticks because they can also serve as disease-carrying vectors that can affect the health of other farm animals [36,52]. Nevertheless, the role of antibiotics in the treatment of TBDs [53] and the influence of environmental factors on tick prevalence have received limited attention [54]. The economic impact of ticks and TBDs among grazing animals remains largely unknown, despite indications that heavy tick infections can significantly affect animal health and lead to substantial annual economic losses in livestock productivity, especially in developing countries.

Farmers generally support the use of anti-tick vaccines, and this attitude may be linked to their experiences of vaccination during the COVID-19 pandemic. However, such vaccines are not currently available in the region, and a study showed that farmers were least aware of the anti-tick vaccine in this region [9]. One epidemiological study from Pakistan confirmed the frequent use of acaricides for controlling tick burden on farm animals [55]. To effectively address the issue of TBDs, the development of an effective anti-tick vaccine is essential, as it could prove to be more efficient and cost-effective than acaricides [56,57]. Although respondents expressed concerns about ticks and TBDs, this was reported negatively due to their use of multiple irregular sprays to minimize the spread of ticks. This behavior contributes to the development of resistance among ticks, leading to heavy tick burdens on animals [35].

During the summer, farmers move their animals to mountain pastures. Moving to summer pastures introduces significant changes for the animals. For example, cows may experience shifts in their diet, which can lead to variations in digestion and nutrient absorption when they move to summer pastures [58]. This season also creates ideal conditions for the growth and spread of ticks, increasing the possibility that animals may become infested and develop tick-borne diseases (TBDs) [59]. If the affected animals are not provided with adequate feed, there may be a high chance of mortality among grazing animals. A positive attitude was noted among farmers, as they typically provide medical treatment to sick animals. A study from this region also reported that most respondents contacted a veterinary doctor when their animals were sick [60]. Pakistan is located in a subtropical region where tick infestations typically peak during the summer. In this season, ticks are more active because of warm weather [61]. Respondents reported a strategic approach for effective livestock management for treating infected animals medically and separating them from healthy animals. They also ensured that clean water and adequate food were provided to grazing animals, which demonstrates a commitment to animal welfare. However, farmers often sell infected animals that do not recover after medical treatment. This behavior poses a potential risk of infection transmission to other animals and themselves. This attitude suggests a lack of awareness of the long-term risks associated with infection transmission. Farmers should consult veterinary doctors to ensure that any animals they are purchasing or transporting to other locations are free of ticks [62]. Another identified risk factor was the potential financial burden of maintaining infected animals. Many farmers recognize that improper dilution can accelerate the development of acaricide resistance [47]. Moreover, inadequate government subsidies for acaricides exacerbate the problem, possibly leading to increased tick resistance. The government needs to promote a more proactive response at both the national and international levels to address the challenges of implementing tick control programs specifically designed for rural communities [63]. Sustainable tick control strategies that consider factors responsible for acaricide control failures, such as resistance development among ticks and environmental influences are crucial for controlling ticks and TBDs [44].

Most farmers did not tuck their pants into socks or wear dark-colored clothing to protect themselves from tick infestation, which may be attributed to financial constraints [64]. The Department of Public Health website recommends wearing dark-colored clothing and tucking pants into socks as preventive measures [65]. One study reported that dark clothing attracts fewer ticks [66]. Additionally, another study that examined ticks infesting horses in the Balochistan region has also indicated a lack of attention toward tick bites [9]. Our study also emphasizes that most farmers do not recognize specific tick habitats in grazing areas, unknowingly exposing themselves and their livestock to tick-infested areas. Furthermore, they often fail to check their bodies for the presence of ticks when returning home, a behavior that may increase the risk of ticks and TBDs [25,67,68]. Most of them do not bathe upon returning home. However, this behavior has been shown to have no significant effect on removing ticks from the body [46].

The transition from a univariate model to a multivariate model to adjust for values of the variable can provide insight into more protective measures [69]. For example, adjusting values for factors such as clothing choice during fieldwork and performing body inspections for ticks after returning from the field in the multivariable model may help to reduce the spread of ticks and TBDs. This study revealed that multiple factors impact both animals and their farmers due to tick infestation. Recommendations have been proposed to improve these factors. For example, (a) the government should organize tick control training programs at least twice a year, (b) sustainable tick control strategies should be developed and promoted among local people and farmers, (c) cooperation between veterinarians and farmers should be enhanced, (d) the government should allocate substantial funding to support poor farmers financially [70], (e) the knowledge of farming communities should align with veterinary knowledge to reduce the incidence of tick resistance [26], (f) anti-tick vaccine trails must be conducted at the regional level to mitigate the risks associated with ticks and TBDs [71], (g) interaction between veterinary doctors and farmers should be amplified to improve their KAP regarding ticks and TBDs [35], and (h) the role of NGOs is crucial in raising awareness and providing tick-related education to farmers with the collaborative assistance of local communities. These recommendations are also illustrated in Figure 7.

The limitations of our study include the possible influence of our sample size and reliance on the recommendations of local farmer communities. This was due to limited access to the internet and postal services. Another significant reason for the difficulty in accessing all farmers was the challenging geography of the study region, characterized by largely damaged roads. The exclusion of farmers under 18 years of age could be another limitation because this age group may be susceptible to ticks and TBDs. Furthermore, we did not collect data on tick infestation in pet animals because our primary focus was on gathering information regarding small livestock. This aspect is important for addressing zoonosis in animals [30]. 

## 5. Conclusions

Farmers with low knowledge exhibited a higher rate of infection among their livestock, a situation further exacerbated by the indiscriminate use of acaricides. These farmers must combine the traditional knowledge of their rural farming communities with the practical knowledge of veterinary practitioners. Our survey results revealed their limited understanding of control measures to protect themselves from ticks and TBDs. Most farmers did not wear dark colored clothes and tucked pants into socks, indicating they were at risk of tick infestation while working in the fields. Furthermore, respondents should ensure that their children do not interact with animals without confirming that the animals are free of ticks. The role of these three stakeholders (i.e., government, NGOs, and veterinary doctors) was found to be limited, necessitating a more interactive and participatory manner to develop extension programs aimed at educating farmers. Lastly, the farmers expressed a keen interest in updating their knowledge about ticks and TBDs and improving behavior and prevention methods.

## Figures and Tables

**Figure 1 vetsci-11-00497-f001:**
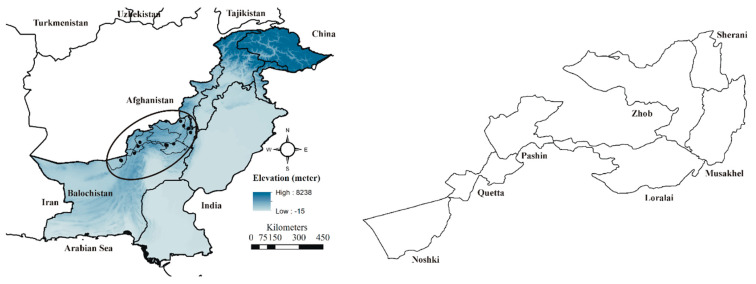
Map of the study area including seven districts of Balochistan (map was designed using ArcGIS version 10).

**Figure 2 vetsci-11-00497-f002:**
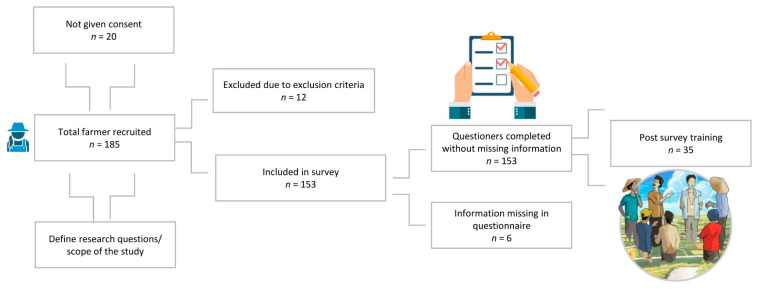
General overview of participants.

**Figure 3 vetsci-11-00497-f003:**
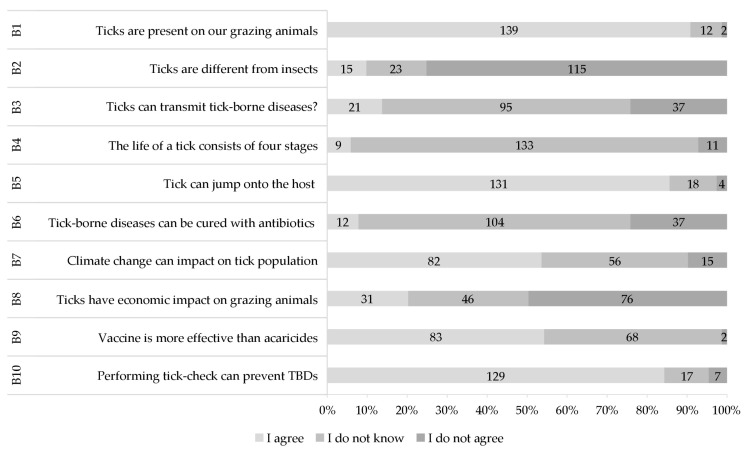
Knowledge-based results in the context of ticks and tick-borne diseases (all numbers in %).

**Figure 4 vetsci-11-00497-f004:**
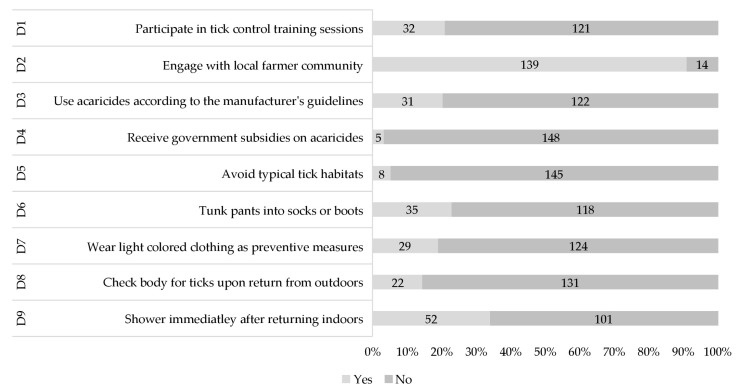
Attitude-based results in the context of ticks and tick-borne diseases (all numbers in %).

**Figure 5 vetsci-11-00497-f005:**
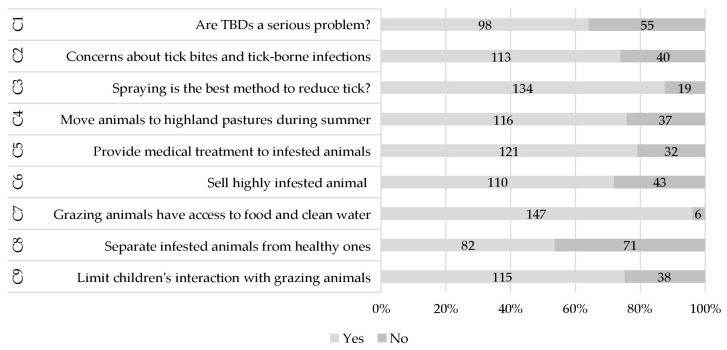
Practice-based results in the context of ticks and tick-borne diseases (all numbers in %).

**Figure 6 vetsci-11-00497-f006:**
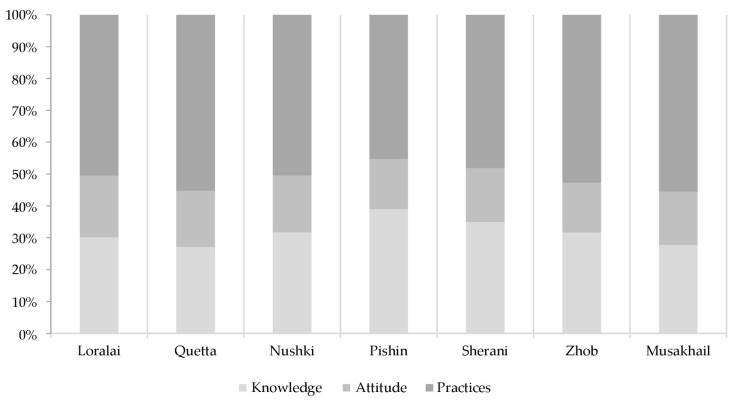
Percentage distribution scores of knowledge, attitude, and practices (KAP) survey in each district.

**Figure 7 vetsci-11-00497-f007:**
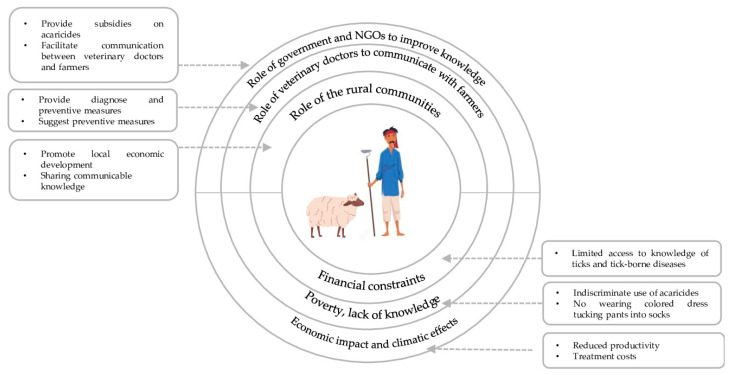
The interconnected roles of key stakeholders, i.e., government, NGOs, veterinary doctors, and rural communities, in enhancing knowledge and practices related to tick management and tick-borne diseases.

**Table 1 vetsci-11-00497-t001:** Outcomes and their substantial solution used in the study.

Q. No.	Response Action	Suggested Action	Source of Information
B1	Agree: Confirm the presence of ticks	Important for risk understanding	[30]
B2	Agree: Ticks are arachnids, not insects	Essential for education	[35]
B3	Agree: Awareness of ticks and TBDs transmission	Fundamental knowledge about ticks and TBDs	[36]
B4	Agree: Understanding tick life cycle	Useful for control strategies	[37]
B5	Agree: Ticks attached through contact	Address misconceptions	[37]
B6	Agree: Antibiotics can treat some ticks and TBDs	Comprehensive treatment is required	[38]
B7	Agree: Climate change impacts tick population	Useful for risk prediction	[39]
B8	Agree: Economic impact on grazing animals due to tick infestation	Important for cost–benefit analysis	[39]
B9	Agree: Availability of vaccine	Emphasis on current limitations	[40]
B10	Agree: Regular check tick presence	A practical approach for tick management	[36]
C1	Yes: Concern about ticks and TBDs	Awareness of impact	[41]
C2	Yes: Concerns about the tick bites and Tick-borne infection	Awareness level	[41]
C3	Yes: Preference to use spraying on infested animals	Reliance on acaricides	[35]
C4	Yes: Seasonal migration of animals	Supports control strategies	[39]
C5	Yes: Provide medical treatment to ill animals	Important for treatment	[9]
C6	Yes: Sell infected animals	Affects welfare	[39]
C7	Yes: Provide food and water	Good husbandry practices	[36]
C8	Yes: Separately place infected animals	Critical for control	[42]
C9	Yes: Keep children away from infected animals	Awareness of zoonotic risks	[30]
D1	Yes: Education enhances tick management	Supports effective control	[30]
D2	Yes: Collaboration improves strategies	Encourages community approaches	[43]
D3	Yes: Proper acaricidal use is crucial	Ensures safety and efficacy	[44]
D4	Yes: Financial support from the government	Supports economic control	[37]
D5	Yes: Awareness reduces tick exposure	Useful for integrated management of ticks	[41]
D6	Yes: Protective measures prevent bites	Practical prevention in the field	[45]
D7	Yes: Preventive clothing helps spot ticks	Enhances protection	[35]
D8	Yes: Personal tick checks after the field	Essential for protection	[45]
D9	Yes: Showering to minimize tick attachment	Ineffective prevention	[46]

**Table 2 vetsci-11-00497-t002:** Socio-demographic characteristics of farmers (*n* = 153).

Questions	Variable	Frequency	Percentage (%)	*p*-Value
Age	18–24	41	26.8	0.04
	25–45	65	42.48	
	45–60	36	23.53	
	>60	11	7.19	
Marital status	Single	32	14.38	0.33
	Married	121	85.62	
Ethnicity	Pashtoon	86	54.86	0.13
	Baloch	52	23.97	
	Others	15	8.58	
Urbanicity	Urban	18	11.76	0.41
	Rural	135	88.24	
Qualification	Illiterate	26	16.99	0.04
	Primary education	61	39.87	
	Secondary education	55	35.95	
	College and above	11	7.19	
Experience in dealing with animals (years)	<5	21	13.73	0.03
	5–10	19	12.42	
	10–15	55	35.95	
	>15	58	37.91	
Average monthly family income USD (USD 1 = PKR 290)	>200	34	23.13	0.18
	200–300	101	68.24	
	>300	18	8.63	
Type of grazing animals	Goats	1721	59.52	0.16
	Sheep	921	31.85	
	Cattle	119	4.12	
	Buffalo	45	1.56	
	Camels	0	0	
	Horses	3	0.10	
Pet animals	Dog	72	86.75	0.34
	Cat	11	13.25	
	Rabbit	0	0	

**Table 3 vetsci-11-00497-t003:** Binary logistic ordinal regression model of associated risk factors.

Factors	Univariant Model		Multivariant Model	
	Unadjusted OR 95% CL	*p*-Value	Adjusted OR 95% CL	*p*-Value
Dark-colored clothing choice	0.1584 (0.04–056)	0.00	-	-
Community engagement	0.17 (0.05–0.53)	0.00	-	-
Body inspection	1.14 (0.4–2.75)	0.82	0.58 (0.24–1.37)	0.29
Visits to tick habitat	1.02 (0.19–5.31)	0.65	0.20 (0.01–0.05)	0.31
Indiscriminate use of acaricides	0.37 (0.16–0.84)	0.02	-	-
Impact of climate change	9.57 (4.54–20.19)	0.00	0.75 (0.38–1.48)	0.49

## Data Availability

All data presented in this study are available in the manuscript.

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
