# Peer review of "Unveiling Misconceptions among Small-Scale Farmers Regarding Ticks and Tick-Borne Diseases in Balochistan, Pakistan"

_vetsci, 2024, doi:10.3390/vetsci11100497_

Round 1

Reviewer 1 Report

Comments and Suggestions for Authors

The manuscript is about KAP survey in small livestock-holders regarding Ticks and tick-borne diseases in Balochistan, Pakistan.

Kindly add some results in the abstract. Results are general in the abstract.

Line 63-64: Infectious diseases including chicken pox, scabies, measles, tuberculosis, and leishmaniasis pose significant challenges among livestock farmers of Balochistan. Kindly add tick-borne diseases examples.

It would be informative if you could add number of tick species and tick-borne diseases present in Balochistan. The current KAP study has significance to devise tick management strategies to prevent tick-borne diseases. 

Methodology is ok.

Results are well presented.

Discussion: There is no comparison of this KAP survey results with other similar studies conducted in Pakistan. For example, authors mentioned in the introduction that "In Pakistan, only a few KAP studies have been conducted which are not sufficient in determining the level of knowledge and practices among farmers [19,21,22]". How current study has/have similarity or differences with other studies?

Conclusion is ok.

Comments on the Quality of English Language

Minor editing is required.

Author Response

Author: additional results has been incorporated.

Line 63-64: Infectious diseases including chicken pox, scabies, measles, tuberculosis, and leishmaniasis pose significant challenges among livestock farmers of Balochistan. Kindly add tick-borne diseases examples.

Author: Examples of TBDs has been given next to the infectious diseases reported from Balochistan.

It would be informative if you could add number of tick species and tick-borne diseases present in Balochistan. The current KAP study has significance to devise tick management strategies to prevent tick-borne diseases.

Author: In the introduction section, we have included reported studies on tick-borne diseases and provided the names of the most prevalent species responsible for their spread.

Discussion: There is no comparison of this KAP survey results with other similar studies conducted in Pakistan. For example, authors mentioned in the introduction that "In Pakistan, only a few KAP studies have been conducted which are not sufficient in determining the level of knowledge and practices among farmers [19,21,22]". How current study has/have similarity or differences with other studies?

Author: All relevant literature available in Pakistan has been incorporated into the discussion section and compared with the findings of this study.

Reviewer 2 Report

Comments and Suggestions for Authors

Generally, the manuscript has interesting findings relevant to farmers, research community and policy makers and other relevant stakeholders. However It can be improved by addressing some of the comments I have below

1. Title

The term "small livestock-holders" is misleading as it might imply small-sized animals (small ruminants) or  small-scale farmers. Specify.

2. Simple Summary and Abstract

Align Simple Summary and Abstract more closely in their message e.g. . Simple Summary states that farmers lack awareness about the economic effects of ticks, while the Abstract mentions "a moderate level of knowledge."

Also better to mention the specific location of the study (Balochistan province, Pakistan) the Simple Summary, , as in the Abstract.

“Awareness and education programs" is repeated multiple times in both the Simple Summary and Abstract. Consider rewording or consolidating these statements to avoid redundancy.

In the Simple Summary, link lack of awareness to the neglect of preventive measures to strengthen the argument. Similarly, in the Abstract, the point about neglecting preventive measures (lines 35–37) should be more strongly linked to the lack of awareness discussed earlier.

Clarify what "proper" acaricide use and "coloured clothing" entail to provide more specific guidance.

Elaborate on the specific actions stakeholders can take to address TTBDs

Methods.

Regarding the sample size, provide more information on how the 185 farmers were 125

recruited for the survey. Sample size calculation?

Results

The statement in the results “The results (Line 242-244) of the logistic regression model using the Fisher exact test showed that 242 no clothing choice during grazing hours and indiscriminate use of acaricides contrary to 243 manufacturer guidelines significantly affected tick infestation rates”. Please attempt to explain this relationship in the discussion section.

Result presentation may need improvement. Somehow looks dull.

Author Response

  1. Title

The term "small livestock-holders" is misleading as it might imply small-sized animals (small ruminants) or small-scale farmers. Specify.

Author: Changes has been made in the title.

  1. Simple Summary and Abstract

Align Simple Summary and Abstract more closely in their message e.g.. Simple Summary states that farmers lack awareness about the economic effects of ticks, while the Abstract mentions "a moderate level of knowledge."

Also better to mention the specific location of the study (Balochistan province, Pakistan) the Simple Summary, as in the Abstract.

"Awareness and education programs" is repeated multiple times in both the Simple Summary and Abstract. Consider rewording or consolidating these statements to avoid redundancy.

In the Simple Summary, link lack of awareness to the neglect of preventive measures to strengthen the argument. Similarly, in the Abstract, the point about neglecting preventive measures (lines 35-37) should be more strongly linked to the lack of awareness discussed earlier.

Author: Abstract is further improved, and simple summary and abstract details are cross-checked and recommended sentences have been improved.

Clarify what "proper" acaricide use and "colored clothing" entail to provide more specific guidance. Elaborate on the specific actions’ stakeholders can take to address TTBDs.

Author: The role of stakeholders, along with the proper use of acaricides and the impact of wearing colored clothing on ticks, has been addressed in the discussion section.

Methods.

Regarding the sample size, provide more information on how the 185 farmers were 125 recruited for the survey. Sample size calculation?

Author: A statement is already provided in the material and method section i.e., This study was based on random clusters of farmers recommended by the local farmer community, because their data is not registered with the Animal Husbandry Department of Balochistan, Pakistan.

Results

The statement in the results "The results (Line 242-244) of the logistic regression model using the Fisher exact test showed that 242 no clothing choice during grazing hours and indiscriminate use of acaricides contrary to 243 manufacturer guidelines significantly affected tick infestation rates". Please attempt to explain this relationship in the discussion section.

Result presentation may need improvement. Somehow looks dull.

Author: The Results section has been improved and revised. The results of the logistic regression model are properly explained regarding these factors i.e., clothing choice and indiscriminate use of acaricides and compared with previously reported studies in the Discussion section.

Reviewer 3 Report

Comments and Suggestions for Authors

Line 21, 26, 29, 38, 42, 43, 66, 93, 94, 98, 101, 117: not "TTBDs", but "TBD" - tick-borne diseases

Line: 69: CCHF (??)

Line 146, 147, 148 and Table 1.: TBD

Line 188, 194, 196, 223, 231, 251, 253, 269, 270, 279, 285, 302, 309, 315, 323: TBD

Author Response

Comments and Suggestions for Authors Line 21, 26, 29, 38, 42, 43, 66, 93, 94, 98, 101, 117: not "TTBDs", but "TBD" - tick-borne diseases

Line: 69: CCHF (??)

Line 146, 147, 148 and Table 1.: TBD

Line 188, 194, 196, 223, 231, 251, 253, 269, 270, 279, 285, 302, 309, 315, 323: TBD.

Author: Corrections are made as per suggestion.

Reviewer 4 Report

Comments and Suggestions for Authors

The topic of the paper is very interesting and the issue is important for animal and human health. The target group is crucial for risk prevention.  However, there are several points of criticism in the presentation of the study, which I found described in an unclear and inconsistent manner in several parts. 

Introduction

Line 66 and line 72: I would expect some information on the main tick-borne zoonoses and epidemiological data on the specific area of the study (see line 74)

Line 74 why the parenthesis? Revise the sentence. Here, since it has been said that tick-borne diseases are rare I expect to read some epidemiological data. 

Lines 83-85. The sentence should be rephrased. It is not clear to me how KAP studies can provide solutions to workers' risk exposure by increasing their knowledge and rectifying misconceptions. Usually, KAP analyses capture self-reported knowledge, attitudes, and practices, providing insights into the target group. Bridging the gap between knowledge and actual behaviour is one of the crucial issues in psychosocial studies and risk communication. I would expect more literature on this and on social studies on TTBD.

Lines 96-102 - The research objectives: apart from point i) the others (ii, iii) are not presented and developed in the paper. This part needs to be revised, the objectives must be consistent with the results presented.

Materials and methods 

The method is not clearly explained and the description provided does not provide a satisfactory understanding of how the study was carried out.  In particular, there is no detailed description of how the questionnaire was administered. 

Lines 137 141. Information on the sample are mentioned, so I would include them in section 2.3 Study sample. 

Line 137.  There is mention of a stratified sample (samples??) but the stratification variables are not made explicit. 

Line 139. Unclear, in what sense were the respondents encouraged to circulate the survey? Line 131 mentions face-to-face interviews.

Line 152. What does it mean “solution” concerning the questions in the questionnaire? Clarify the meaning of the term in the context of use. If values have been assigned these must be made explicit in relation to each item.

Line 156 – The motivation is not clear. Explain the data collection procedure in more detail. 

Table 1. The meaning of the table is not clear. It should be clarified and specified in the text what is listed in the OUTCOME column (what Agree, do not agree, and disagree mean. Also if agree and do not agree are the same thing, it should be standardised). SOLUTION Also the solution column is unclear as stated above. It is difficult for me to understand the meaning or usefulness of these labels.  Moreover,  I find inconsistencies with figures 3 and 4. E.g.: B9 - from figure 3 more than availability seem to be efficacy. Also among some outcomes of C part attitude, they have unclear formulations or do not seem congruent with the item in the figure of results. 

Section 2.5. Statistical analysis: not very detailed description, in particular, it is not indicated how the scores for the three KAP parameters are calculated. If scores were assigned based on the answers indicated as right or wrong, they should be specified. The values assigned to each item considered for the calculation of the score as well as for the attitude and practice items should also be declared.

Results 

Generally speaking, there are considerations that are not anchored to the data and relationships between the items that cannot be deduced from the data presented, therefore it is not clear whether they are hypotheses or comparisons with the literature that are not made explicit (e.g. Lines 206-207 it is not clear from the data presented the consequentiality declared in the text ‘IF any animals does not recover after medication, THEY OFTEN PREFER TO SELL the infested animal in the market. From the table, I see two separate items, no preference for selling infested animals who did not recover could be inferred. If the relationship is hypothesized it should be specified; also in Line 220 all cause-effect relationships described do not seem supported by data...etc.) 

Figures 3/4/6 It is unclear how the items were presented and formulated in the questionnaire. Figure 4 Some items are in question form others are not. It would be important to provide at least in the appendix/supplementary the precise wording of the questions presented to the farmers 

Figure 4. Many items seem to me to be farming practices. In what sense are they considered attitudes? 

Figure 5. It is unclear the sense of its inclusion in the paper. If other parts of research have been developed involving observation of animals and identification of the presence of ticks, and the authors want to include them in this paper, they should also be detailed in the materials and methods and results sections and then discussed. I cannot find that, so I would suggest omitting this part. 

Line 231. I would expect to find the brochure in methods.

Discussion and conclusion

It is not clear to me the distinction between the results of the study and the comments and results from the literature. Interpretations of the results in several parts do not seem to be supported by the data presented. If other insights were made or open questions were presented during the survey that allow quantitative data to be supplemented with motivations/explanations it should be made explicit and detailed in the methods and results (e.g. line 259 that cats are taken to entertain children is not on the results; Line 274 how do they say that they consider vaccine a feasible alternative BECAUSE they have experienced vaccination during Covid? Even in the Conclusion section line 332 Owing to financial constraints, most of them were not wearing long shoes and colorful clothes, I cannot find data to support this relation between the clothes and poverty. In the absence of data clearly showing the causal relationship or the reasons given for interpreting the data, the researchers' interpretations need to be supported by arguments and literatures. 

Figure 8. The first and the third boxes on the left have the same contents referring, it seems to me, to different categories. Is this correct? 

A limitation part should be inserted.

Author Response

Introduction

Line 66 and line 72: I would expect some information on the main tick-borne zoonoses and epidemiological data on the specific area of the study (see line 74).

Author: Few additional lines have been incorporated to further strengthen the argument regarding the presence of tick-borne zoonoses and the available literature on ticks from the study province.

Line 74 why the parenthesis? Revise the sentence. Here, since it has been said that tick-borne diseases are rare I expect to read some epidemiological data.

Author: The sentence given in parentheses has been removed and updated.

Lines 83-85. The sentence should be rephrased. It is not clear to me how KAP studies can provide solutions to workers' risk exposure by increasing their knowledge and rectifying misconceptions. Usually, KAP analyses capture self-reported knowledge, attitudes, and practices, providing insights into the target group. Bridging the gap between knowledge and actual behaviour is one of the crucial issues in psychosocial studies and risk communication. I would expect more literature on this and on social studies on TTBD.

Author: Thank you for highlighting the importance of these points. Additional lines with supportive references have been added to the sections mentioned by the reviewer. Furthermore, local study references have been included to provide an overview of the region.

Lines 96-102 - The research objectives: apart from point i) the others (ii, iii) are not presented and developed in the paper. This part needs to be revised, the objectives must be consistent with the results presented.

Author: For clarity of the readers objectives has been revised and focused only on KAP related parameter.

Materials and methods

The method is not clearly explained, and the description provided does not provide a satisfactory understanding of how the study was carried out. In particular, there is no detailed description of how the questionnaire was administered.

Author: Thank you for highlighting this important aspect of the questionnaire that was not previously addressed in the original manuscript. We have now included additional details explaining how the questionnaire was designed and refined. These changes have been incorporated into the revised manuscript.

Lines 137 141. Information on the sample are mentioned, so I would include them in section 2.3 Study sample.

Author: The mentioned lines have been moved to the study sample section.

Line 137. There is mention of a stratified sample (samples??) but the stratification variables are not made explicit.

Author: The term 'stratified sample' has been removed to avoid confusion and ensure clarity in the revised manuscript.

Line 139. Unclear, in what sense were the respondents encouraged to circulate the survey? Line 131 mentions face-to-face interviews.

Author:  In line 139, we have added a statement indicating that we encouraged respondents to circulate the survey to maximize responses from farmers. Additionally, we conducted face-to-face in-person interviews to ensure comprehensive engagement with those respondents who lack access to the internet and email. For more clarity face-to-face word is replaced with in-person.

Line 152. What does it mean "solution" concerning the questions in the questionnaire? Clarify the meaning of the term in the context of use. If values have been assigned these must be made explicit in relation to each item.

Author: The word solution is replaced with recommended approach.

Line 156 - The motivation is not clear. Explain the data collection procedure in more detail.

Author: In response to your suggestion, we have provided a detailed explanation of the data collection procedure within the same paragraph.

Table 1. The meaning of the table is not clear. It should be clarified and specified in the text what is listed in the OUTCOME column (what Agree, do not agree, and disagree mean. Also if agree and do not agree are the same thing, it should be standardised). SOLUTION Also the solution column is unclear as stated above. It is difficult for me to understand the meaning or usefulness of these labels. Moreover, I find inconsistencies with figures 3 and 4. E.g.: B9 - from figure 3 more than availability seem to be efficacy. Also among some outcomes of C part attitude, they have unclear formulations or do not seem congruent with the item in the figure of results.

Author: We are thankful for suggestion to further improve Table 1. The labels in Table 1, previously titled "Outcome" and "Solution," have been replaced with "response action" and "suggested action." A statement has been incorporated into the main text to strengthen the adoption of this table in the manuscript: "This table provides a structured guide to understanding respondents' knowledge, attitudes, and practices related to ticks and TBDs, as reported in previous studies, to further enhance KAP-related activities on their farms for better livestock management." Further, the content of Table 1 is carefully revised and cross-checked with the questions of KAP study and special emphasized is given on C part of the results.

Section 2.5. Statistical analysis: not very detailed description, in particular, it is not indicated how the scores for the three KAP parameters are calculated. If scores were assigned based on the answers indicated as right or wrong, they should be specified. The values assigned to each item considered for the calculation of the score as well as for the attitude and practice items should also be declared.

Author: Statistical analysis has been revised and updated. A clear statement has been provided regarding the attitudes and practices, explaining how the data were analyzed.

Results

Generally speaking, there are considerations that are not anchored to the data and relationships between the items that cannot be deduced from the data presented, therefore it is not clear whether they are hypotheses or comparisons with the literature that are not made explicit (e.g. Lines 206-207 it is not clear from the data presented the consequentiality declared in the text 'IF any animals does not recover after medication, THEY OFTEN PREFER TO SELL the infested animal in the market. From the table, I see two separate items, no preference for selling infested animals who did not recover could be inferred. If the relationship is hypothesized it should be specified; also in Line 220 all cause-effect relationships described do not seem supported by data...etc.)

Author: We appreciate the careful examination of our manuscript. We have thoroughly cross-checked the results, particularly the attitude section. Corrections have been made in ‘Practices section’ specially in the mentioned lines, i.e., 206-207 and 220. Additional statements in these lines, which could cause confusion regarding whether the data is presented with or without a hypothesis, have been removed.

Figures 3/4/6 It is unclear how the items were presented and formulated in the questionnaire. Figure 4 Some items are in question form others are not. It would be important to provide at least in the appendix/supplementary the precise wording of the questions presented to the farmers.

Author: a brief version is given in the Figures whereas the detailed questions asked to participants is provided in the questionnaire as supplementary material.

Figure 4. Many items seem to me to be farming practices. In what sense are they considered attitudes?

Author: We have carefully revised the interpretation of results to further define the concept of 'attitude' in relation to KAP studies (please the result of attitude).

Figure 5. It is unclear the sense of its inclusion in the paper. If other parts of research have been developed involving observation of animals and identification of the presence of ticks, and the authors want to include them in this paper, they should also be detailed in the materials and methods and results sections and then discussed. I cannot find that, so I would suggest omitting this part.

Author: We agree with the reviewer's remarks, and therefore Figure 5 and its caption have been removed from the main text.

Line 231. I would expect to find the brochure in methods.

Author: The brochure has been moved to the Methods section.

Discussion and conclusion

It is not clear to me the distinction between the results of the study and the comments and results from the literature. Interpretations of the results in several parts do not seem to be supported by the data presented. If other insights were made or open questions were presented during the survey that allow quantitative data to be supplemented with motivations/explanations it should be made explicit and detailed in the methods and results (e.g. line 259 that cats are taken to entertain children is not on the results; Line 274 how do they say that they consider vaccine a feasible alternative BECAUSE they have experienced vaccination during Covid? Even in the Conclusion section line 332 Owing to financial constraints, most of them were not wearing long shoes and colorful clothes, I cannot find data to support this relation between the clothes and poverty. In the absence of data clearly showing the causal relationship or the reasons given for interpreting the data, the researchers' interpretations need to be supported by arguments and literatures.

Author: Sentences in the Discussion were modified to clearly differentiate between current study results and comments from literature. Moreover, the information, including “…cats are taken to entertain children…” and “…have experienced vaccination during Covid…” are not our results but comments from the literature and our speculation, used to suggest possible reasons why farmers have cats and why they support anti-tick vaccination, respectively. Similarly, the finding that farmers do not wear long shoes and colorful clothes is based on our results, for which possible reasons, including financial constraints, were cited from the literature in the Discussion; this reason was removed from the Conclusion in the revised manuscript.

Figure 8. The first and the third boxes on the left have the same contents referring, it seems to me, to different categories. Is this correct?

Author: Thank you for pointing the mistake. The corrections have been made in the revised version of the manuscript.

A limitation part should be inserted.

Author: The limitation related to our study is provided in the last paragraph of the discussion section.

Round 2

Reviewer 4 Report

Comments and Suggestions for Authors

The authors improved the text and explained some aspects. In my opinion, there are still some critical issues regarding the questionnaire. The items are not very clear and if it was sent and filled out independently by the farmers, it may have been misinterpreted. If it was administered face to face, I deduce that, as specified by the authors in the text, it was also explained to the interviewees, given their level of education. I suppose that the items are so essential to be as simple as possible in relation to the level of education of the target. However, these aspects need to be clarified.

The data analysis still lacks details on the construction of the KAP indices.

Study sample

It remains unclear whether the questionnaire was administered in mixed mode, online and face to face or only in face-to-face mode to all respondents.

If it was also administered in online mode, it should be specified whether it was computerized through some software and how respondents’ consent was collected if there was no direct contact with the breeders. The formulation of the questions as they are presented in the supplementary material does not seem to me to be suitable for self-compilation.

If both online and face-to-face questionnaires were collected, the number of respondents for the two data collection methods must be indicated.

Table 1 is still not clear to me, because I see all Agree and Yes marked, and I interpret that Agree and Yes are considered the correct answers in the case of knowledge and the protective answer in the case of attitude and practices, however, I understand that not all items are positive eg.

-          C6. Sell highly infested animals - the answer YES is it considered a protective attitude?

-          D7. Wear light-colored clothing - the answer YES is it considered a protective practice?

If there are items in which the “correct” answer should be NO or I do not agree, it must be specified.

The link between Table 1 and the data analysis is not clear, because in relation to the YES and AGREE I would expect to find a guide to the calculation of the KAP score (see comment in statistical analysis). But this is not explained.

Statistical analysis

It is not clear how the KAP parameter score presented in figure 6 is calculated. It is necessary that the score assigned to each item in the questionnaire is clear (see comment on table 1)

Results

Line 258 - "A strong belief was reported regarding moving animals to highland pastures during summer, as respondents linked better weather and food availability in the mountainous areas with improved animal health (n = 116, 75.81%)".

Where does it come from that farmers link better weather and food availability to the mountainous areas? See also line 338 of the discussion

Discussion

Line 306 "However, their living expenses were very low, which forced them to dilute the acaricides”

Where is it stated that they dilute the acaricides? From the questionnaire we can deduce that they do not use them according to the manufacturer's guidelines, however, we do not know how they use them and if the incorrect use is linked to income…

Lines 337 - 339 The data does not support the motivations related to moving to the mountains, nor that they are aware that summer is the season of the greatest expansion of ticks

Line 355 The sentence is not clear.

Line 365 It is not clear the meaning of long shoes.

Line 374. In what sense then can item D9 "having a shower after returning indoors" be evaluated as protective? How is it considered in the calculation of the score of protective behaviors if it is declared that has no influence?

Figure 7. Please check again the text in the boxes and the correct association with stakeholders or obstacles to preventive measures. Please also revise the caption of the figure.

Conclusion

Line 407- The article does not present epidemiological data. The association between knowledge and the actual level of infestation of animals is not supported.

Line 417 – Also the level of interest in updating knowledge I don't think it can be understood from the questionnaire.

Author Response

______________________________________________________________________

Editor

Veterinary Science, MDPI

Dear Editor,

The manuscript “Unveiling Misconception among Small Livestock-Holders Regarding Ticks and Tick-Borne Diseases in Balochistan, Pakistan” was revised to “Unveiling Misconception among Small-Scale Farmers Regarding Ticks and Tick-Borne Diseases in Balochistan, Pakistan” and the suggestions were added in the revised manuscript. We greatly appreciate your time for handling our submission and assessing the review comments. The issues raised have been addressed in the revised MS.

Sincerely,

Abid Ali, PhD

REVIEWER-5th

Comments and Suggestions for Authors

The manuscript is about KAP survey in small livestock-holders regarding Ticks and tick-borne diseases in Balochistan, Pakistan. The authors improved the text and explained some aspects. In my opinion, there are still some critical issues regarding the questionnaire. The items are not very clear and if it was sent and filled out independently by the farmers, it may have been misinterpreted. If it was administered face to face, I deduce that, as specified by the authors in the text, it was also explained to the interviewees, given their level of education. I suppose that the items are so essential to be as simple as possible in relation to the level of education of the target. However, these aspects need to be clarified.

The data analysis still lacks details on the construction of the KAP indices.

Author: Thank you for all your efforts on our submission. The materials and method section has been revised, highlighting that administering the questionnaire online or through postal services was not possible. Therefore, we conducted the questionnaire face to face. The detail of the corrected paragraph which is revised in the materials and methods section. The use of email and internet facilities is either limited or not accessible to farmers in the study region. All questions were administered via in-person interviews to ensure a high response rate and accurate data collection”.

Moreover, the statistical analysis under the materials and methods section has also been revised for clarity.

Study sample

It remains unclear whether the questionnaire was administered in mixed mode, online and face to face or only in face-to-face mode to all respondents.

Author: The questionnaire was administered solely through face-to-face interviews. We have clarified this by specifying “face to face interviews” in the manuscript.

If it was also administered in online mode, it should be specified whether it was computerized through some software and how respondents’ consent was collected if there was no direct contact with the breeders. The formulation of the questions as they are presented in the supplementary material does not seem to me to be suitable for self-compilation.

Author: We have clarified in the revised manuscript that the questionnaire was administered in print form and shared with the study participants, not online. The supplementary material, including the questionnaire in Pakistan’s national language, Urdu, has been provided. Additional details regarding the questionnaire’s administration have been included in the revised manuscript, in response to your first comment.

If both online and face-to-face questionnaires were collected, the number of respondents for the two data collection methods must be indicated.

Methodology is ok.

Author: Only face-to-face questionnaires were conducted, as mentioned in the revised manuscript. The administration of online questionnaires was not possible due to the geographic landscape of the study area and the limited availability of resources such as internet access and postal services.

Table 1 is still not clear to me, because I see all Agree and Yes marked, and I interpret that Agree and Yes are considered the correct answers in the case of knowledge and the protective answer in the case of attitude and practices, however, I understand that not all items are positive eg.

Results are well presented.

Author: To improve clarity, additional lines were added to the materials and methods (Survey Instrument-Questionnaire Section). We have clarified that the “response action” represents the expected answers, such as “Yes” or “I agree” from farmers, and “suggested actions” provide recommendations for controlling ticks and tick-borne diseases (TBDs), based on insights from published articles.

 C6. Sell highly infested animals - the answer YES is it considered a protective attitude?

Author: For C6 (Table 1): “Sell highly infested animals”, the answer “YES” indicates that it will affect animal welfare. In response to your comments, details on the “response action” (“Yes” or “I agree”) and “suggested actions” have been provided in the revised manuscript.

D7. Wear light-colored clothing - the answer YES is it considered a protective practice?

Author: For D7 (Table 1): “Wear light-colored clothing”, the answer “YES” indicates that it enhances protection. In response to your comments, details on the “response action” (“Yes” or “I agree”) and “suggested actions” have been provided in the revised manuscript.

If there are items in which the “correct” answer should be NO or I do not agree, it must be specified.

Author: In Table 1, all answers are either “YES” or “AGREE”. For clarity, the table is intended to serve as a reference for understanding the KAP study. There are no items for which the “correct” answer should be “NO” or “I do not agree”.

The link between Table 1 and the data analysis is not clear, because in relation to the YES and AGREE I would expect to find a guide to the calculation of the KAP score (see comment in statistical analysis). But this is not explained.

Author: A detail for KAP score is given in the material methods in the mentioned paragraph ‘Knowledge-based questions (n = 10) were assessed based on three primary categories: 'I agree,' 'I disagree,' and 'I don't know'. The response option i.e., ‘I agree’ means that the respondent confirms the knowledge statement provided in the question, ‘I do not know’ indicates a lack of knowledge regarding the statement, and ‘I do not agree’ means that the respondent does not agree with the statement. In contrast, ‘Yes’ or ‘No’ options were provided in the attitude and practices sections to simplify the response of respondents because these sections focus on specific behaviors, which makes the binary choice more appropriate’.

Table 1 is further explained in lines 173-176. Hope this will provide sufficient information why this table was designed and included in the manuscript.

Statistical analysis

It is not clear how the KAP parameter score presented in figure 6 is calculated. It is necessary that the score assigned to each item in the questionnaire is clear (see comment on table 1)

Author: A supplementary table has been provided for the KAP calculation, where a score of 1 is assigned for answers of 'YES' or 'I Agree' in the KAP questionnaire. Additional details on the evaluation and scoring process are also provided in the Materials and Methods section (Statistical Analysis section). For more information, you can download the Excel sheet by copying and pasting the link below:

https://docs.google.com/spreadsheets/d/1ivCFho6p1GAd8uoDHk8dvvLXGII4FZw/edit?usp=sharing&ouid=101788729409476664657&rtpof=true&sd=true

Results

Line 258 - "A strong belief was reported regarding moving animals to highland pastures during summer, as respondents linked better weather and food availability in the mountainous areas with improved animal health (n = 116, 75.81%)". Where does it come from that farmers link better weather and food availability to the mountainous areas? See also line 338 of the discussion

Author: It was the response of the farmers reported in the attitude section of the questionnaire. Discussion regarding mentioned paragraph has been revised and improved.

Discussion

Line 306 "However, their living expenses were very low, which forced them to dilute the acaricides”

Where is it stated that they dilute the acaricides? From the questionnaire we can deduce that they do not use them according to the manufacturer's guidelines, however, we do not know how they use them and if the incorrect use is linked to income…

Author: sentence has been deleted.  

Lines 337 - 339 The data does not support the motivations related to moving to the mountains, nor that they are aware that summer is the season of the greatest expansion of ticks

Author: Corrected the concept in the discussion section.

Line 355 The sentence is not clear.

Author: Revised the sentence.

Line 365 It is not clear the meaning of long shoes.

Author: The concept of long shoes has been replaced, and the original concept of tucking pants into shoes is provided in the mentioned lines

Line 374. In what sense then can item D9 "having a shower after returning indoors" be evaluated as protective? How is it considered in the calculation of the score of protective behaviors if it is declared that has no influence?

Author: This question was asked to verify whether farmers take a bath after spending the whole day on the farms. Although the removal of ticks after bathing has been reported to have no significant effect, as mentioned in the following article, this same article is also referenced in the discussion section.

Schimpf, D.J.; Ewert, M.M.; Lai, V.K.; Clarke, B.L. Responses of ticks to immersion in hot bathing water: Effect of surface type, water temperature, and soap on tick motor control. PLOS ONE 2021, 16, e0261592.

Figure 7. Please check again the text in the boxes and the correct association with stakeholders or obstacles to preventive measures. Please also revise the caption of the figure.

Author: Corrected and updated the Figure 7.

Conclusion

Line 407- The article does not present epidemiological data. The association between knowledge and the actual level of infestation of animals is not supported.

Author: Corrected the sentence in the mentioned line.

Line 417 – Also the level of interest in updating knowledge I don't think it can be understood from the questionnaire.

Author: This sentence is revised in the conclusion section.